# Surgical Outcomes of Laminectomy, Durotomy and a Non-Synthetic Dura Substitute Application in Ten Dogs with a Spinal Subarachnoid Diverticulum

**DOI:** 10.3390/vetsci11030128

**Published:** 2024-03-14

**Authors:** Michał Mól, Ricardo Fernandes, Simon Wheeler, Massimo Mariscoli

**Affiliations:** 1Paragon Veterinary Referrals, Part of Linnaeus, Paragon Business Village Paragon Way, Red Hall Cres, Wakefield WF1 2DF, UK; ricardo.fernandes@blaise-referrals.com (R.F.); simon.wheeler@me.com (S.W.); massimo.mariscoli@paragonreferrals.co.uk (M.M.); 2Blaise Referrals, 1601 Bristol Road South, Longbridge, Birmingham B45 9UA, UK

**Keywords:** veterinary neurosurgery, subarachnoid diverticulum, dog, laminectomy, durotomy, a non-synthetic dura substitute application

## Abstract

**Simple Summary:**

Spinal subarachnoid diverticula are fluid dilations in the subarachnoid space that can cause compression of the spinal cord and associated neurological dysfunction such as paresis, spinal ataxia, and urinary and faecal incontinence. This study describes the signalment, clinical signs, diagnostic imaging findings, surgical treatment, and outcomes of laminectomy, durotomy, and the application of a non-synthetic dura substitute in ten dogs with a spinal subarachnoid diverticulum. Based on the good long-term postoperative results, we concluded that dura substitute application is a viable and safe technique in dogs for the surgical treatment of subarachnoid diverticulum.

**Abstract:**

This retrospective study aimed to report the surgical treatment and outcomes of laminectomies followed by durotomy and the application of a non-synthetic collagen matrix dura substitute (Durepair^TM^) in ten dogs with a spinal subarachnoid diverticulum (SAD). The medical records of these ten client-owned dogs with SAD diagnosed by magnetic resonance imaging (MRI) were reviewed. All patients had chronic and progressive deficits. At presentation, common neurological signs were proprioceptive ataxia, ambulatory spastic paraparesis or tetraparesis, and faecal incontinence. Dorsal thoracolumbar laminectomy was performed in eight dogs; one dog underwent cervical dorsolateral laminectomy, and one patient had thoracic hemilaminectomy. Laminectomies were followed by durotomy, allowing the dissection of the pia-arachnoid adhesions. A rectangular patch of a non-synthetic dura substitute was applied as an onlay graft over the durotomy site before routine closure. Proprioceptive ataxia, paraparesis, and tetraparesis improved in all patients. Faecal incontinence in one patient resolved postoperatively. Laminectomy, durotomy, and the application of a non-synthetic dura substitute was a safe procedure facilitating postoperative improvement over a long-term follow-up period (from 9 to 40 months).

## 1. Introduction

Spinal subarachnoid diverticulas (SADs) are fluid dilations in the subarachnoid space that can cause the compression of the spinal cord and associated myelopathy. Multiple publications describe this condition using a variation of cyst denotations: leptomeningeal cyst [1], meningeal cyst [2], spinal meningeal cyst [3], subarachnoid dilatation [4], spinal arachnoid pseudocyst [5], spinal intradural arachnoid cysts [6], and spinal subarachnoid cysts [7]. Histologically, this “cystic”-type lesion has no epithelial cell lining; therefore, inaccurate “cyst” terminology has recently been replaced by the term subarachnoid diverticulum. SADs have been described in people [8], dogs [1,2,3,4,5,6,7,9,10,11,12,13,14,15,16,17,18,19,20,21,22,23,24,25], cats [26,27], and one horse [28].

SADs can occur anywhere in the vertebral column; cervical lesions are most common in large breeds, whilst thoracolumbar lesions are more prevalent in small breed dogs. Pugs, French Bulldogs, and Rottweilers are the most affected breeds, as reported in a large study on 122 dogs from three different institutions [9].

Various aetiologies have been suggested in dogs, including hereditary and congenital, biomechanical factors, and concurrent or previous spinal disorders such as intervertebral disc extrusion and protrusion, vertebral malformation, spinal instability [15], or meningo-myelitis [9].

The “typical” clinical signs include mild to severe, often progressive pelvic limb or generalised proprioceptive ataxia and paresis not associated with spinal hyperaesthesia, paresis, and faecal and/or urinary incontinence.

MRI is an imaging modality of choice for the diagnosis of SAD [11,22], given its greater detail in evaluating spinal cord parenchyma, the anatomic location of the cystic mass, its size, and its relation to neuronal structures.

Most of the veterinary data about SADs originate from small case series or case reports. This hampers the further understanding of breed and sex predispositions, age distribution, variations in presenting clinical signs, and the prevalence of concurrent and possibly associated spinal abnormalities. Surgical treatment is superior to medical treatment in the management of SAD in dogs [12]. Many reports use different surgical techniques and have short follow-up times, making the estimation of the long-term benefits of treatment challenging. In dogs, the recurrence of neurological deficits has been described in a small number of cases between 5 and 44 months postoperatively [22,25].

Dura mater can be damaged in SAD cases as a result of a primary lesion and from surgical intervention. Over several decades, a variety of materials have been used as grafts, ranging from rubber sheeting in 1895 to fill the dural defect [29] to various synthetic polymer sheets [30,31,32,33]. These materials fell out of favour due to high infection rates, their inability to integrate with the dura, foreign body encapsulation, a high rate of adhesion formation, and epidural and subdural hematomas [34,35]. Collagen-based dural graft substitutes, typically derived from animal collagens, processed to remove immunogenic components, replaced those dural substitutes with poor biological performance.

We report the application of Durepair (Durepair Regeneration Matrix, Medtronic, Minneapolis, MN, USA), a strong collagen graft of intact dermis from foetal bovine, processed to remove all cellular components and not chemically modified with crosslinking chemicals [36]. This native collagen graft does not elicit an inflammatory response, is immunologically accepted, reconstituted with host cells, and supports vasculature and remodelled [36]. Durepair is straightforward to handle, strong, economical, and well tolerated by the dogs when incorporated into the recipient site [36].

The aim of this retrospective study was to describe the signalment, history, clinical signs, and diagnostic imaging findings in a cohort of 10 patients using an identical surgical technique, laminectomy including durotomy and the application of a non-synthetic dura substitute, and long-term postoperative follow-up to determine the clinical outcome. 

## 2. Material and Methods

The medical records of dogs with chronic, progressive paraparesis, or tetraparesis and ataxia diagnosed with SAD and surgically treated at the authors’ institution between September 2020 and September 2022 were retrospectively reviewed. Only dogs with complete medical records, a general physical and neurological examination, spinal cord compression caused by the single teardrop-shaped dilatation of the subarachnoid space reported on an MRI and surgically confirmed and treated with an identical surgical technique, and a minimum postoperative follow-up time of 9 months were included. Two Pugs, with presumptive constrictive myelopathy associated with a constrictive circumferential band compressing the spinal cord, a focal subarachnoid space irregular margination reported on MRI and treated with a matching surgical technique, were excluded from the study. Additionally, three French Bulldogs with SAD treated with durotomy and Durepair application combined with spinal stabilisation were also excluded.

SAD was identified on an MRI in all cases, and in 5 cases, CT was also performed. SAD was confirmed surgically in all dogs. The neurological status was scored using the modified Frankel Scale [37]: grade 0 (neurologically normal), grade 1 (spinal pain without neurological deficits), grade 2 (ambulatory paraparetic/ataxic), grade 3 (non-ambulatory paraparetic), grade 4 (paraplegia with intact nociception), or grade 5 (paraplegia with absent nociception). The urinary and faecal continence status was recorded. The spinal MRI was performed in all cases with a 1.5 T permanent magnet unit (Toshiba, Otawara, Japan). The MRI protocol included sagittal, dorsal, and transverse T1- and T2-weighted images, transverse fluid-attenuated inversion recovery (FLAIR), sagittal and transverse T1 post gadolinium, and sagittal fast advanced spine echo (FASE) in all dogs. Five patients also underwent CT scanning using a 16-slice unit (Toshiba, Otawara, Japan). A CT was performed to assess and characterise the presence of vertebral malformation and/or caudal articular processes dysplasia. The imaging studies were interpreted and reported by board-certified radiologists.

Surgical outcomes were divided into short-term (up to 4 weeks postoperatively), mid-term (more than 4 weeks and less than 9 months), and long-term (9 months or more). Follow-up assessments were performed with neurological examinations or by telephone consultations and gait analysis by video footage provided by the owners at standardised time points. The owner’s satisfaction level of the neurological outcome was recorded each time. The outcome was considered as positive if there was an improvement in the gait, defecation status, or combination of the above-mentioned. The outcome was deemed negative if these neurological signs deteriorated or there was evidence of a relapse of clinical signs. If the neurological status was unchanged, the outcome was defined as stable. Postoperative minor complications were defined as self-limiting or medically manageable. Major complications were defined as life-threatening circumstances requiring urgent surgical or medical management.

## 3. Results

Ten dogs fulfilled the inclusion criteria as follows: four French Bulldogs, three Pugs, one Rottweiler, one Staffordshire Bull Terrier, and one West Highland White Terrier. The youngest dog was 5 months old at the time of onset of neurological signs, and the oldest was 93 months. The median age was 41 months (mean 44.1 months). The research group consisted of 6 male and 4 female dogs. The median weight was 10.5 kg (mean 10.5; range 6.5–35.0 kg). 

### 3.1. History and Clinical Signs

The duration of neurological deficits at presentation ranged from 1 month to 24 months, with a median of 2 months (mean: 5.2 months). All dogs were referred with a chronic progressive gait abnormality (Table 1). Faecal incontinence was reported in one dog.

The neurological examination was consistent with ambulatory proprioceptive ataxia and spastic paraparesis in nine cases. These patients were localised to the T3–L3 spinal cord segment (SCS) (dogs 1–7, and 9, 10). Dog 8 presented with ambulatory spastic tetraparesis and proprioceptive ataxia with left lateralisation and dysmetria localised to the C1–C5 SCS. Because of the age, history, and neurological signs with the absence of spinal hyperaesthesia, SAD was considered the most likely differential diagnosis.

### 3.2. Preoperative Evaluation

#### 3.2.1. Minimum Data Base

Routine haematology, serum biochemistry, and electrolyte results were within the normal limits in all patients.

#### 3.2.2. Diagnostic Imaging

MRI findings consisted of the teardrop-shaped widening of the dorsal subarachnoid space at the site of the SAD, associated with the compression of the spinal cord in all dogs (Table 2). The lesions were hyperintense on T2-weighted sequences (Figure 1) and iso- or hypointense on T1-weighted images in all cases. FLAIR images showed lesion suppression in seven cases, indicative of CSF accumulation. The sagittal FASE image showed the abrupt attenuation of the affected subarachnoid space and apparent cranial or caudal dilatation of the contiguous subarachnoid space consistent with SAD in all cases. There was no gadolinium enhancement on T1-weighted post-contrast sequences in any MRI studies. SAD was located in the thoracic and thoracolumbar regions from the T7 to L1 level in nine small breed dogs. A large breed dog, Rottweiler, was diagnosed with a C2–C3 SAD lesion. T13–L1 was the most affected site (3/10), followed by T10–T11 (2/10), T12–T13 (2/10), T9–T10 (1/10), T7 (1/10), and C2–C3 (1/10).

CT revealed concurrent spinal or vertebral disorders at the same level or near the diverticulum in five dogs (Table 2). These spinal deformities included the following: the aplasia or hypoplasia of articular processes (n = 3), intervertebral disc protrusion (n = 1), kyphosis (n = 1), and ventral spondylosis deformans (n = 1). The Staffordshire Bull Terrier was diagnosed with acute non-compressive nucleus pulposus extrusion (n = 1) 14 months before, at the level where SAD later developed.

### 3.3. Anaesthesia, Analgesia, and Perioperative Period

All patients were anaesthetised twice as follows: first for diagnostic imaging and later for surgical procedures. Premedication included a combination of methadone 0.2–0.3 mg/kg and medetomidine 5 µg/kg intravenously (IV) in all cases. General anaesthesia was induced with alfaxalone 2–3 mg/kg IV and maintained with isoflurane in medical-grade oxygen. Ketamine was administered intraoperatively at the 0.5 mg/kg IV loading bolus followed by 10 µg/kg/min to reduce inhalant anaesthetic requirements through an additive or synergistic action and to produce additional perioperative analgesia [38]. The continuous intravenous infusion of ketamine was maintained for 24 to 48 h after surgery at a 2 µg/kg/min dose to ensure adequate pain control. A broad-spectrum antibiotic (cephalexin 22 mg/kg every 1.5 h IV) was administered intraoperatively, and gastroprotectant omeprazole (1 mg/kg) was administered preoperatively and continued for two days postoperatively in brachycephalic dogs.

Postoperative analgesia was selected for each individual dog involving methadone, and/or buprenorphine and ketamine in accordance with the Glasgow modified pain score. The dogs were hospitalised until injectable analgesic discontinuation; good pain control was achieved with oral medications, and adequate movement in the limbs and voluntary urination was observed. Postoperative analgesia also included a combination of gabapentin (10 mg/kg every 8 h for 14–21 days), meloxicam (0.2 mg/kg IV once, followed by 0.1 mg/kg orally every 24 h for 7–14 days), or robenacoxib (1–2 mg/kg every 24 h for 5–10 days), and paracetamol (10 mg/kg every 8 h for 5–10 days) at the attending clinician discretion.

### 3.4. Surgical Management

Dorsal thoraco-lumbar laminectomy was performed in eight dogs; one dog underwent right-sided cervical dorso-lateral laminectomy, and one patient had a caudal thoracic hemilaminectomy. All dogs were positioned in sternal recumbency. A midline incision was performed in each case, followed by the sharp and blunt dissection of the paravertebral muscles until the vertebral column was exposed (Figure 2). Before starting the laminectomies and prior to entering the vertebral canal, a neurosurgical operating microscope (Zeiss OPMI CS NC-2, Germany) was used to improve the visualisation of the lamina, dura, spinal cord, and anomalous tissue. A durotomy was performed with the initial stab incision of the meninges using a disposable ophthalmic keratome straight blade (Alcon Knife, Denmark) at the level of the SAD. When CSF flowed out, a longitudinal durotomy with Castroviejo scissors was extended in both the cranial and caudal direction until normal spinal cord parenchyma was exposed. Stay sutures on both sides of the dural edges were then applied using polydioxanone 6-0 (PDS II Ethicon LLC, San Lorenzo, Puerto Rico) and maintained in place with haemostats. The identified adhesions were surgically removed using dura micro dissector instruments to allow the careful repositioning of the neural tissue (Figure 3). Following the resolution of SAD adhesions, a rectangular-shaped patch of non-synthetic dura substitute Durepair was trimmed. The size of the graft selected was based on the size of the laminectomy, allowing appropriate placement on the dural defect with sufficient overlap. The patch was rehydrated prior to its use for 5 min by soaking it in a 0.9% sterile saline solution, and it then was applied onlay onto the durotomy site, slightly extending over the margins without suturing to the surrounding structures. The muscular, subcutaneous, and skin layers were closed routinely.

No perioperative complications were reported. All dogs had an uneventful recovery from surgery and anaesthesia.

### 3.5. Follow-Up and Outcome

Minor postoperative complications included subcutaneous seroma in dog 9. The seroma resolved after 2 weeks without the need for aspiration or surgical treatment. Major postoperative complications were unobserved.

The initial worsening of paraparesis and tetraparesis was reported in 50% of the cases (Table 2). Dogs were discharged from the hospital between 2 and 7 days after surgery.

The short-term follow-up was available in all patients at the 3-week postoperative appointments. The initial postoperative deterioration of tetraparesis and paraparesis was followed by a gradual gait improvement in all cases.

The mid-term follow-up (from 1 month to 7 months, with a median of 1.4 months and a mean of 2.4 months) was available in eight patients. All patients were ambulatory with gradual gait improvement. Complete faecal continence was obtained in dog 3.

The long-term follow-up (ranging from 9 months to 40 months, with a median of 27.5 months, mean: 26 months) was available in all dogs. All dogs showed significant gait improvement during 9-month postoperative rechecks compared with their preoperative neurological status. Dog 4, however, returned to a pre-surgical status 18 months postoperatively, showing moderate spastic ambulatory paraparesis and pelvic limb ataxia. The owner declined advanced imaging, and long-term physiotherapy was elected. This dog had the same neurological deficits—moderate spastic ambulatory paraparesis—32 months after the surgery. Dog 6 was euthanised 10 months postoperatively due to unrelated causes. The same patient had no neurological deficits during the 9-month postoperative recheck. Dog 2, with no neurological deficits at the 16-month postoperative follow-ups, developed the recurrence of mild spastic ambulatory paraparesis 26 months postoperatively. The owner declined advanced diagnostic imaging, and long-term physiotherapy was elected. The last recheck of this patient took place 30 months postoperatively. The dog was ambulatory with mild spastic paraparesis, which was still better than prior to the SAD procedure when paraparesis was moderate. All owners were satisfied with the neurological status of their dogs during long-term follow-ups. No dog had faecal incontinence. Telephone or in-person questionnaires obtained 11 months or later after the procedures in 9 cases reported a 100 percent satisfaction rate in 8 cases and a 90% satisfaction rate in dog 8 in regards to the patient’s neurological status.

Overall, we found laminectomy, durotomy, and this non-synthetic dura substitute application surgical technique to be safe, not cause an abnormal inflammatory response, and effective in healing the surgically created defects in the dura mater.

## 4. Discussion

This retrospective study describes the favourable outcomes of surgically treated dogs with SAD where a dural substitute (Durapair) was applied. To the authors’ knowledge, this is the first study reporting the clinical results of onlay Durapair application in the spinal cord of dogs. The follow-up outcomes suggest that a collagen-based dural substitute is safe in dogs affected by SAD and treated surgically via a laminectomy and durotomy.

In a canine model, following implantation, Durepair was populated by fibroblasts and blood vessels. Device resorption led to cell penetration, vascularisation, and collagen remodelling. The mechanical properties of Durepair resembled native dura in a canine duroplasty study [36]; the material was well tolerated by the animals, and no adhesions were seen at 3 months. However, at 6 months, a limited number of focal adhesions were noted at the healing suture margins where Durepair edges were sutured to create intracranial durectomy sites. These represented an inflammatory reaction to the suture material and were not of clinical importance. In our cases, Durepair was not sutured to the dura, which the authors considered to be a safe way to overcome suture inflammatory reactions. Moreover, CSF leakage may be seen through the needle holes themselves despite adequate suture closure in human dural surgical repair technique research [39]. Therefore, we recommend Durepair to be applied over the dural defect without suturing. The onlay application of Durepair might also help in overcoming the potential herniation of the spinal cord through the laminectomy site.

Clinical studies in human neurosurgery have found dural grafts made of bovine pericardium to be well tolerated. In the reports consisting of 31, 32, and 22 patients who underwent xenogeneic pericardium duroplasty, outcomes were good or excellent, whereas fair or poor outcomes were seen in 3 patients not related to surgical closure [40,41,42]. One case report described an allergic reaction to Durepair in a woman following spinal cord untethering, which was successfully resolved after graft removal [43].

The surgical outcomes of dogs treated for SAD and a comparison between different surgical methods and conservative treatments are scarce in the veterinary literature. The medical SAD treatment aims to decrease CSF production, lower its volume, and reduce perilesional inflammation [11]. Medical treatment ranges from exercise restrictions to tapering courses of prednisolone at anti-inflammatory doses [11]. Although prednisolone is often the medication of choice for reducing CSF production to treat dogs with fluid accumulation in the subarachnoid space, there is no evidence of its efficacy for that condition [44]. Medications such as prednisolone possibly reduce secondary perilesional oedema, but this does not provide a resolution in clinical signs. Omeprazole, a proton pump inhibitor, has been proposed as an adjunctive treatment for reducing CSF production. However, a study found that CSF production in healthy dogs might not be affected by chronic oral therapy with omeprazole [45].

One study compared the outcomes of medical management to surgical treatment in 96 dogs [12]. In total, 50 dogs were managed medically, and 46 were treated surgically. Of the 38 dogs treated surgically with available long-term follow-ups, 82% of dogs improved, 3% remained stable, and 16% deteriorated after surgery. Of the medically treated dogs with available long-term follow-up, 30% improved, 30% remained stable, and 40% deteriorated. Surgical treatment was more often associated with clinical improvement compared to medical management in this study.

Surgical treatment focuses on the decompression of the spinal cord for the improvement of clinical signs. Various SAD procedures and approaches are reported depending on the lesion localisation and surgeon’s preference, including hemilaminecomy, dorsal laminectomy, and ventral slot. Variation in durotomy [23], durectomy [24], diverticulum fenestration [13], dural marsupialisation [14,18,19,23], fenestration with the closure of the durotomy site [20] and placing a shunt tube to restore normal CSF flow [21,23], with [19,23] or without vertebral stabilisation are described. An effective definitive therapy has yet to be determined. Despite the numerous surgical options available, the aim of this surgery was to dissect leptomeningeal adhesions at the time of surgery. However, the veterinary literature lacks information about protective and/or regenerative measures for the diseased meninges.

In a study on 13 dogs of various breeds suffering from SAD, 66% of dogs had postsurgical improvement (8/13), with the follow-up period ranging from 6 to 30 months [7]. In other studies describing outcomes after the surgical treatment of SAD with follow-up carried from 6 months to 2.5 years postoperatively, there was an improvement in 66% (eight of the twelve dogs) [10] and an 82% improvement in thirty-eight surgically treated dogs with a median follow-up of 23 months [12]. The median time to relapse of the neurological deficits was 20.5 months after different SAD surgical techniques in a study of seven dogs and one cat with a median follow-up time of 36 months [22].

All our cases showed long-term (>9 months) improvement of neurological signs. Long-term follow-up showed the complete (n = 7) or partial resolution (n = 3) of the neurological abnormalities. This favourable outcome might have been the result of durotomy with the surgical release of the diverticulum and the fibrotic tissue compressing the spinal cord, followed by Durepair application. The use of surgical microscopy is reported to enable surgeons to carefully detach adhesions between meninges, the spinal cord, and nerves, with improved surgical outcomes [46]. The sutureless concept of dural repair reduces the surgical time and facilitates patch placement in an anatomically challenging location. Dural substitute application was selected to restore the normal anatomy of the meninges and spinal cord parenchyma, eliminating abnormal CSF leakage and further accumulation with subsequent spinal cord tissue compression. This should prevent the further impingement of the nervous tissue, which we speculate to be the main contributor to our successful outcomes.

A control, postoperative MRI, would have been useful for confirming or excluding the above-mentioned hypothesis. Postoperative MRI was not performed on any dog included in the study as this cannot be justified either ethically or economically in the majority of cases.

The breeds and the neurolocalisation recorded in our study reflected the most frequently reported clinical signs and breed-related lesion localisation in the largest published retrospective study of SAD [9]. Paresis, hypermetria, and ataxia depending on the area of the spinal cord involved, were the presented neurological deficits. Paresis and proprioceptive ataxia are presumably caused by an impairment of the dorsally located ascending proprioceptive pathways, and the hypermetria might be explained by the compression of the spinocerebellar tracts, located dorsolaterally in the spinal cord [12]. Faecal incontinence was present in a dog with thoracolumbar lesions. Upper motor neuron faecal incontinence has reportedly occurred more commonly in SAD patients compared with other spinal cord disorders [12]. It has been suggested that this upper motor neuron faecal incontinence is related to the dorsal compression of sensory pathways that sustain conscious defecation [47]. The disruption of these pathways interrupts the relay of sensory information to the sensory cortex via the thalamic nuclei for the conscious recognition of rectal distension and defecation [11]. All our dogs were presented without spinal hyperaesthesia, corresponding to other authors’ findings [9,10].

There was an association between the body weight and localisation of the SAD and the breed predisposition to certain locations. Small breed dogs were diagnosed with SAD at the T7 to L1 level, and the Rottweiler was diagnosed with a C2–C3 lesion, which is consistent with previous reports [5,7,9]. Rottweilers, a breed with a relatively large head compared to the body, had a cranial cervical SAD predisposition, which supports a previously proposed biomechanical influence attributable to their heavy head [7]. For thoracolumbar SADs, T13-L1 was the most affected site (n = 3), which could result from high spinal mobility in this region [7,10]. In agreement with previous studies, most SADs resulted in a mid-dorsal localisation [5,9,10].

French Bulldogs (n = 4) and Pugs (n = 3) were the most affected breeds, with the overrepresentation of brachycephalic dogs (n = 7) in our study. The number of French bulldogs and Pugs compared with the respective hospital populations reinforces the suspicion of an inherited aetiology, as already observed [9,12].

Previous studies have reported that male dogs are predisposed to SAD [9,12]. In our research, there were 6 males and 4 females. Age seemed not to have an influence on the localisation of the SAD, similar to a study on 122 dogs [9].

The typical MRI findings of SADs appeared as T2W hyperintense, T1W hypointense, and FLAIR hypointense dilations of the subarachnoid space consistent with CSF accumulation, with the characteristic tear-drop shape on sagittal images, as previously described [11]. Not all SADs were hypointense on FLAIR, suggesting that this fluid could be chemically not completely equivalent to CSF in these dogs. FASE images showed abrupt attenuation in the continuity of the affected aspect of the subarachnoid space and the apparent dilatation of the subarachnoid space consistent with an arachnoid diverticulum. This sequence, which is fluid-specific and with a short acquisition time sequence, added value to identify the diverticula of the subarachnoid space, as described in one other study [16].

CT images indicated the presence of the hypoplasia of the caudal articular processes (CAP) in Pugs at the level of the SAD. The use of both MRI and CT in affected Pug dogs to assess the presence of bilateral CAP dysplasia at the level of spinal cord lesions has been advocated in a retrospective study [17]. It is currently unclear why only a minority of dogs with CAP dysplasia develop clinical signs of spinal cord dysfunction, whilst this vertebral anomaly seems to be an incidental finding. It was hypothesised that the high prevalence of CAP dysplasia should not be considered the only factor causing a predisposition to clinical disease in Pugs [48].

Previous or concurrent neurologic diseases in close proximity to the SAD, including intervertebral disc herniation extrusion or protrusion, vertebral malformations, vertebral canal stenosis, fibrocartilaginous embolism, or myelitis of unknown origin, were found to be significantly more common in French Bulldogs and Pugs [48]. Correspondingly, we diagnosed intramedullary lesions, disc protrusions, and vertebral malformations in all French bulldogs.

The retrospective nature of this study focused on the surgical management of SAD, and no comparison with dogs treated conservatively was available. The low number of cases in this study resulted from the low incidence of the disease and, unfortunately, prevents the more objective monitoring of outcomes; definitive conclusions and additional studies are warranted.

## 5. Conclusions

In conclusion, the surgical management of dogs with SAD and dural substitute application led to a significant long-term improvement of the neurological status and was not associated with any major complication or deterioration. Laminectomies, followed by durotomy and dural substitute application, could be considered as a safe and effective treatment of this condition. We believe that the use of a dural substitute covering the dural surgical defect promotes the regeneration of the meninges and spinal cord parenchyma. Durepair application eliminates and minimises the CSF accumulation or leakage, the formation of further adhesions between the surrounding tissues and the meninges and the spinal cord, and prevents secondary spinal cord compression. Effective dural protection may result in improved surgical outcomes. 

## Figures and Tables

**Figure 1 vetsci-11-00128-f001:**
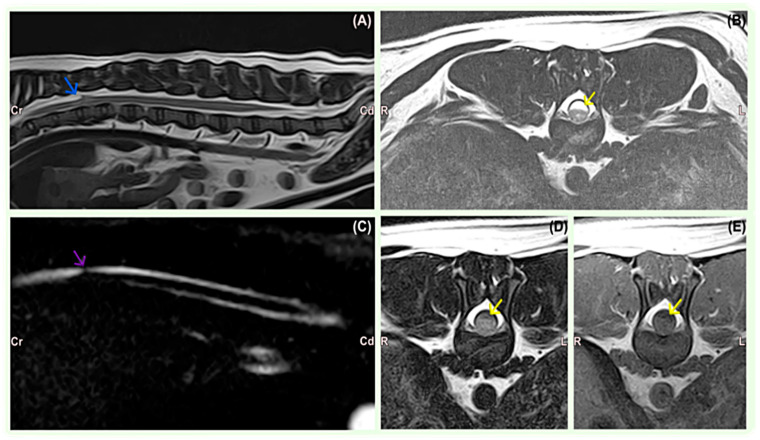
French Bulldog, 0Y8M, male. (**A**) Sagittal T2-weighted image of the thoracolumbar spine showing dilatation of the dorsal subarachnoid space at the level of T10 and T11 (arrow) and the narrowing of the subarachnoid space at the level of the cranial endplate of T12, suggesting leptomeningeal adhesions. (**B**) Transverse T2-weighted image of the spine at the level of T11–T12 showing dilatation of the dorsal subarachnoid space at the level of T10 and T11 (arrow) and a faint increase in the signal intensity of the spinal cord. (**C**) The FASE sequence showed abrupt attenuation in the continuity of the affected aspect of the subarachnoid space (arrow) and apparent dilatation of the subarachnoid space consistent with an arachnoid diverticulum. (**D**) The transverse FLAIR sequence suppressed the lesion, suggesting CSF (arrow). (**E**) Transverse post-contrast T1-weighted image showing no abnormal enhancement identified. The arrow indicates hypointense subarachnoid fluid dilation.

**Figure 2 vetsci-11-00128-f002:**
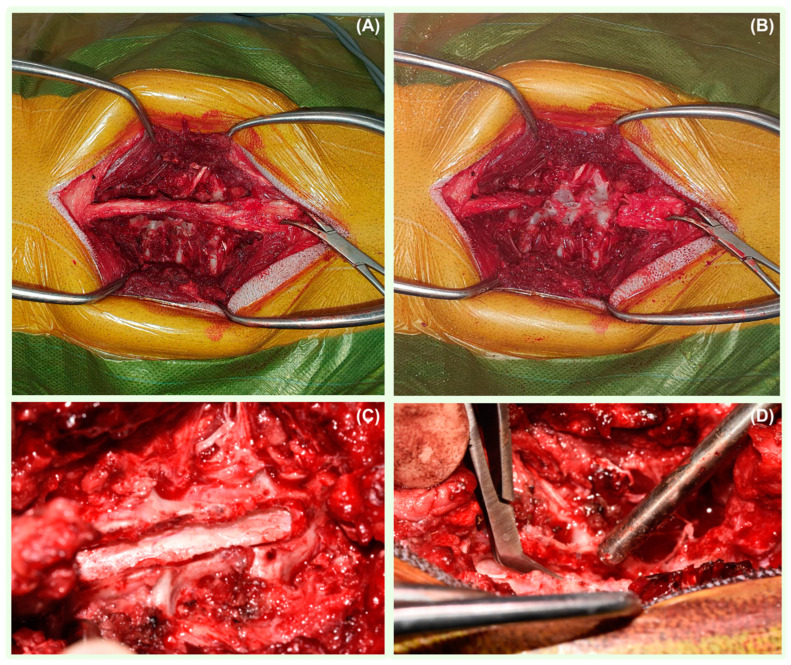
Intraoperative photo demonstrating removed epaxial muscles exposing spinal processes (**A**). The spinal processes were removed (**B**). The completed dorsal laminectomy (**C**). A durotomy was performed (**D**).

**Figure 3 vetsci-11-00128-f003:**
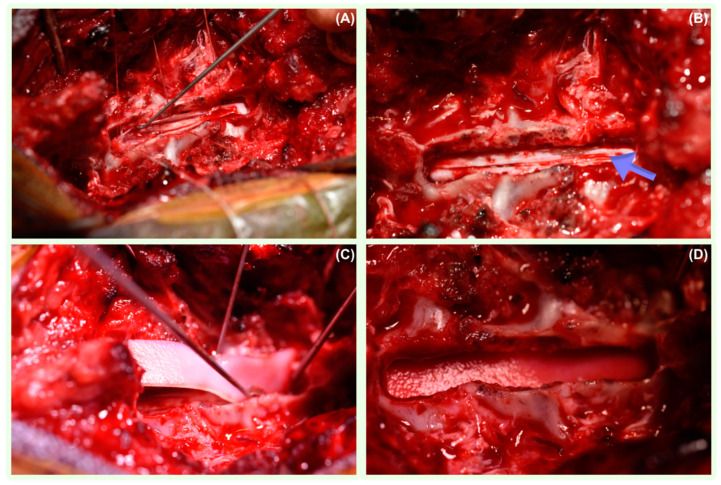
Intraoperative photo demonstrating spinal cord after longitudinal durotomy and removal of the spinal subarachnoid adherences (**A**). Longitudinal durotomy (arrow) following adherence removal. The dural edges subsequently re-joined (**B**). Placement of the non-synthetic dura substitute patch onto the durotomy site (**C**). The dura substitute patch applied onto the durotomy site without suturing it to the surrounding structures (**D**).

**Table 1 vetsci-11-00128-t001:** Demographics, symptoms duration, and neurological signs in 10 dogs with SAD.

Case	Symptoms Duration (Months)	Gait at Presentation	Faecal Incontinence	Urinary Incontinence
1. French Bulldog, 0Y8M M, 12.5 kg	1	Moderate paraparesis, spinal PL(s) ataxia	No	No
2. Pug, 6Y3M FN, 6.9 kg	3	Moderate paraparesis L-lateralisation, spinal PL(s) ataxia	No	No
3. Pug, 8Y11M FN, 6.5 kg	2	Non-ambulatory paraparesis, spinal PL(s) ataxia, faecal incontinence	Yes	No
4. Staffordshire Bull Terrier, 4Y7M F, 14.1 kg	14	Moderate paraparesis, spinal PL(s) ataxia	No	No
5. Pug, 8Y1M M, 10.1 kg	1	Moderate paraparesis, spinal PL(s) ataxia	No	No
6. French Bulldog, 4Y1M MN, 15.7 kg	24	Moderate paraparesis, spinal PL(s) ataxia	No	No
7. French Bulldog, 0Y7M M, 8.6 kg	2	Moderate paraparesis, spinal PL(s) ataxia	No	No
8. Rottweiler, 4Y0M M, 35 kg	1	Moderate ambulatory tetraparesis, L-hypermetria, general proprioceptive ataxia	No	No
9.West Highland White Terrier, 1Y3M F, 8.2 kg	2	Mild paraparesis L-lateralisation, spinal PL(s) ataxia	No	No
10. French Bulldog, 3Y7M, M, 10.9 kg	2	Mild paraparesis, spinal PL(s) ataxia	No	No

**Table 2 vetsci-11-00128-t002:** Diagnostic imaging findings, surgical treatment, and outcome in 10 dogs with SAD.

Case	MRI	CT	Surgery	Immediate Postop Results	Outcome
1. French Bulldog	Dorsal SAD at the T10–T11. A mild intramedullary T2 hyperintensity at the T11–T12	No	Dorsal laminectomy T10–T12	Nonambulatory paraparesis for 3 days, regained ambulation on day 4.	Improved: 30 months postop: no deficits.
2. Pug	Dorsal SAD T13–L1, syrinx T12–T13 Non-compressive T13–L1, L1–L2, L2–L3 disc protrusions	R CAPs T12, T13 atrophy	Dorsal laminectomy T13–L1	Nonambulatory paraparesis for 5 days	Improved: 16 months of no gait deficits, 26 months of mild spastic ambulatory paraparesis recurrence, 30 months mild spastic ambulatory paraparesis.
3. Pug	Dorsal-L-sided SAD T11–T12, mild T11–T12 disc protrusion	T11, T12 caudal CAPs atrophy, ventral spondylosis deformans T10–T13.	Left-sided T11–T12 hemilaminectomy	Nonambulatory paraparetic during 6 days hospitalisation	Improved: 25 months: mild ambulatory paraparesis, no faecal incontinence.
4. Staffordshire Bull Terrier	Dorsal T12–T13 SAD, cranial syringohydromyelia.	Mild thoracolumbar vertebral spondylosis	Dorsal laminectomy T12–T13	Ambulatory paraparesis	Improved: 9 months: mild paraparesis, mild PL(s) ataxia, 32 months: moderate paraparesis.
5. Pug	Dorsal SAD T13–L1	T11, T12, T12, T13, L1 CAPs dysplasia	Dorsal laminectomy T13–L1	Nonambulatory paraparesis for 4 days	Improved: 22 months no PL(s) weakness or ataxia.
6. French Bulldog	Dorsal R-sided SAD T9–T10, hemivertebrae T10, T13	No	Dorsal laminectomy T9–T1	Ambulatory paraparetic	Improved: 9 months no deficits. Euthanised 10 months postop due to acute paraplegia from a traumatic incident
7. French Bulldog	Dorsal SAD T10. T2W-intramedullary hyperintensity T10–T13 secondary syringohydromyelia	Vertebral malformations T7–T12 without spinal cord compression	Dorsal laminectomy T9–T10	Nonambulatory paraparesis for 6 days	Improved: 40 months no deficits.
8. Rottweiler	Dorsal SAD C2–C3 with marked spinal cord compression	No	Right-sided C2–C3 cervical dorso-lateral laminectomy	Nonambulatory tetraparesis for one day	Improved: 32 months: no deficits.
9.West Highland White Terrier	Dorsal SAD T12–T13	No	Dorsal laminectomy T11–T13	Ambulatory paraparetic	Improved: 24 months: no deficits.
10. French Bulldog	Dorsal SAD T7, changes at T8 suggestive of oedema, hemivertebrae T5, T7, non-compressive disc protrusions at C3–C4, C4–C5, T10–T11, T11–T12, L2–L3, L7–S1	No	Dorsal laminectomy T6–T7	Ambulatory paraparetic	Improved: 16 months: no deficits.

Abbreviations: CAPs, caudal articular processes; F, female; L, left; M, male; N, neutered; PL(s) pelvic limb(s); postop, postoperative/-ly; R, right.

## Data Availability

The data that support the findings of this research are available from the corresponding author upon reasonable request.

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
