# Peer review of "Surgical Outcomes of Laminectomy, Durotomy and a Non-Synthetic Dura Substitute Application in Ten Dogs with a Spinal Subarachnoid Diverticulum"

_vetsci, 2024, doi:10.3390/vetsci11030128_

Round 1
Reviewer 1 Report
Comments and Suggestions for Authors
General comments
On the whole this is a well-described report on a series of dogs with subarachnoid diverticula and treated by one specific surgical method. The results are reported with sufficient detail although they could be more clearly presented. The results are encouraging and the claim that this technique is safe is reasonable, although both suppositions must be considered somewhat tenuous based on the small number of cases. The intra-operative photographs are OK but not of great quality / detail and perhaps could be omitted. The table content and presentation could be improved. The discussion would benefit from being more concise.
Specific comments (by line number)
88-91: this is really straining the definition of a hypothesis – would suggest deleting.
94: there is a need to be a little more precise with the definition of SAD here … as you point out, there are many different types / ages / duration of clinical signs at which dogs develop ‘SAD’ which may include ‘scarring’ or lesions associated with disc herniations, articular process anomalies and it would help if the lesion in each case were to be categorized a little further. The cases you report here are clearly not a homogenous group of a single etiology. It would help readers to interpret the value of this technique to have this additional detail.
96: how many cases were excluded because of not fulfilling these requirements? (it’s important because readers need to know how representative of the whole population the reported cases are). There is a wide age range and range of symptom duration too … suggesting a mixture of sub-types.
Table 1: case 6 why was dog euthanased? More detail is required.
Immediate post op results would be better in a table – it is difficult to read as it is.
333: best to leave out 'significant' if the data have not been statistically tested (which would be totally unnecessary / inappropriate for this small number of cases).
343: claims of disproportionality should only be made if you supply data (in the results) to support this statement.
347/369: reword to avoid implication of statistical testing
388: ‘only one research …’ needs rewording
Comments on the Quality of English Language
There are a few places where the language could be improved
Author Response
For research article
Surgical Outcomes of Laminectomy, Durotomy and a
Non-Synthetic Dura Substitute Application in Ten Dogs with a Spinal Subarachnoid Diverticulum
|
Our Response to Comments
|
||||||
|
1. Summary |
|
|
||||
|
Thank you very much for your good comments on our article. We appreciate the time you spent on revision, and we hope that the corrections will meet with approval. Please find the detailed responses below and the corresponding revisions. The corrections, the added ones as well as the crossed-out ones, are highlighted in red in the attached word document. Please see the attachment.
|
||||||
|
2. Questions for General Evaluation |
Reviewer’s Evaluation |
Response and Revisions |
||||
|
Does the introduction provide sufficient background and include all relevant references? |
Yes |
|
||||
|
Are all the cited references relevant to the research? |
Yes |
|
||||
|
Is the research design appropriate? |
Yes |
|
||||
|
Are the methods adequately described? |
Yes |
This is all great! Thank you very much. |
||||
|
Are the results clearly presented? |
Can be improved |
More precise description of the included SAD cases was given in Material and Methods section. The exclusion criteria were added to the article. CSF inclusion was removed for standardization. The table was divided into two parts to make it less bulky and have a space to add immediate post-op results, as per your suggestion. |
||||
|
Are the conclusions supported by the results? |
Can be improved |
We have changed the flow of the article. In the discussion section the primary focus of the study was described 1st changing the lines 376-426 after the lines 317 (according to previous Word file attachment lines numbers). |
||||
|
3. Point-by-point response to Comments and Suggestions for Authors |
||||||
|
Specific comments (by line number): I: 88-91: this is really straining the definition of a hypothesis – would suggest deleting. Thank you for this comment. We have removed this paragraph. (Lines 88-91 in the revised attachment - crossed out in red) II. 94: there is a need to be a little more precise with the definition of SAD here … as you point out, there are many different types / ages / duration of clinical signs at which dogs develop ‘SAD’ which may include ‘scarring’ or lesions associated with disc herniations, articular process anomalies and it would help if the lesion in each case were to be categorized a little further. The cases you report here are clearly not a homogenous group of a single etiology. It would help readers to interpret the value of this technique to have this additional detail. We certainly agree with this comment therefore clear description of the included SAD cases was written in lines 94-100.
Medical records of dogs with chronic, progressive paraparesis, or tetraparesis and ataxia diagnosed with SAD, and surgically treated at the authors’ institution between September 2020 and September 2022 were retrospectively reviewed. Only dogs with complete medical records, general physical and neurological examination, spinal cord compression caused by the single teardrop-shaped dilatation of the subarachnoid space reported on MRI and surgically confirmed and treated with identical surgical technique, and a minimum postoperative follow-up time of 9 months were included. III. 96: how many cases were excluded because of not fulfilling these requirements? (it’s important because readers need to know how representative of the whole population the reported cases are). There is a wide age range and range of symptom duration too … suggesting a mixture of sub-types. Thank you for this valuable comment. Making this change improved the quality of Material and Methods section. This can be found on lines 100-105.
Two Pugs, with a presumptive constrictive myelopathy, associated with a constrictive circumferential band compressing the spinal cord, focal subarachnoid space irregular margination reported on MRI and treated with matching surgical technique, were excluded from the study. Additionally, three French Bulldogs with SAD treated with durotomy, Durepair application, combined with spinal stabilisation were also excluded. IV. Table 1: case 6 why was dog euthanased? More detail is required. Immediate post op results would be better in a table – it is difficult to read as it is.
Thank you for pointing this out. The line was added to table 2. In outcome section. Immediate post op results were added to table 2.
V. 333: best to leave out 'significant' if the data have not been statistically tested (which would be totally unnecessary / inappropriate for this small number of cases). We strongly agree, and the word significant was deleted (line 397)
VI. 343: claims of disproportionality should only be made if you supply data (in the results) to support this statement. We agree, the word disproportional was removed (line 407).
VII. 347/369: reword to avoid implication of statistical testing We agree and this section was modified removing the words implicating of statistical testing (line 407)
VIII. 388: ‘only one research …’ needs rewording Thank you for this comment. This sentence was also changed (line 341).
|
||||||
Thank you very much and kind regards,
Michal Mol on behalf of the authors

Reviewer 2 Report
Comments and Suggestions for Authors
The work is well written, of great scientific importance.
Author Response
For research article
Surgical Outcomes of Laminectomy, Durotomy and a
Non-Synthetic Dura Substitute Application in Ten Dogs with a Spinal Subarachnoid Diverticulum
|
|
|
|
|
We sincerely thank you for your good and motivating comments on our article. We have made some changes to improve quality of our article. The corrections, the added ones as well as the crossed-out ones, are highlighted in red in the attached word document. Please see the attachment. Please note that in the discussion section the primary focus of the study was edited changing the lines 376-426 (corresponding to the revised article lines 331-385) after the lines 317 according to the previous Word document. We have removed one of the photographs (the one with Durepair and kidney dish). The photographs legends were also edited. The references order and numbers were changed according to the newest article version. Please note that the text was also edited to keep the single similarity index below 5%. Inclusion of CSF was removed from the article, therefore some sentences and sections regarding CSF were removed from the manuscript as crossed out in red (lines 14, 120-126, 198-203, 436-439). Please note that I have added my qualifications after my name (line 5).
Thank you very much and kind regards, Michal Mol on behalf of the authors |
||

Reviewer 3 Report
Comments and Suggestions for Authors
Dear Author,
this is a well-written retrospective study, dealing with an interesting topic. It deserves to be published because it brings new information in veterinary medicine. I have only minor comments.
- The title is relevant for the study, as well as the abstract and the simple summary.
-The introduction is well written with a good background and adequate bibliography.
-The M&M are good, mostly for the protocol standardization, considering the retrospective nature of the study. Maybe the inclusion of the CSF results is unnecessary since in three dogs it was made from cerebello-medullary cistern (so cranial to the lesion) and canine distemper virus was tested in all cases but MRI and clinical images it was supposed to have different differential diagnosis. Please if include CSF results, discuss why it was performed in this way and the limit.
- The results section is good, the table is very useful but too bulky. Maybe the use of abbreviations help in compact it (for example instead of "Case, signalment" put only "Case", instead of "Duration of neurologic abnormalities (months)" put only "Duration symptoms",instead of "Laminectomy-type before durotomy and Durepair application" put only "surgery" and instead of "Neurological Outcome, Recheck time" put only "outcome" ..) or divided the table in 2 parts (Table 1: case, duration, gate abn, incont F, U; Table 2 use only the numbers for the cases, MRI, CT, surgery, outcome)
- In the discussion section please describe first the primary focus of the study, with comparison to medical and others surgery management, including comparison with human medicine. Then discuss the other part. (move the part included between the lines 376-428 after the lines 317)
Author Response
For research article
Surgical Outcomes of Laminectomy, Durotomy and a
Non-Synthetic Dura Substitute Application in Ten Dogs with a Spinal Subarachnoid Diverticulum
|
Thank you for the fantastic comments and kindness. It is truly inspiring to remain persistent in our work. Please find the detailed responses below and the corresponding revisions. The corrections, the added ones as well as the crossed-out ones, are highlighted in red in the attached word document. Please see the attachment. We have removed one of the photographs (the one with Durepair and kidney dish). The photographs legends were also edited. The references order and numbers were changed according to the newest article version. Please note that the text was also edited to keep the single similarity index below 5%. Please note that I have added my qualifications after my name (line 5).
|
Point-by-point response to Comments and Suggestions for Authors
1. - The M&M are good, mostly for the protocol standardization, considering the retrospective nature of the study. Maybe the inclusion of the CSF results is unnecessary since in three dogs it was made from cerebello-medullary cistern (so cranial to the lesion) and canine distemper virus was tested in all cases but MRI and clinical images it was supposed to have different differential diagnosis. Please if include CSF results, discuss why it was performed in this way and the limit.
Thank you for this valuable comment. CSF inclusion was removed for standardization, therefore, some sentences and sections regarding CSF were removed from the manuscript as crossed out in red (lines 14, 120-126, 198-203, 436-439). Making this change improved the quality of the article.
2. - The results section is good, the table is very useful but too bulky. Maybe the use of abbreviations help in compact it (for example instead of "Case, signalment" put only "Case", instead of "Duration of neurologic abnormalities (months)" put only "Duration symptoms",instead of "Laminectomy-type before durotomy and Durepair application" put only "surgery" and instead of "Neurological Outcome, Recheck time" put only "outcome" ..) or divided the table in 2 parts (Table 1: case, duration, gate abn, incont F, U; Table 2 use only the numbers for the cases, MRI, CT, surgery, outcome)
The table was divided into two parts to make it less bulky, as per your suggestion. Moreover, it created a space to add immediate post-op results. Use of abbreviations was applied.
3. - In the discussion section please describe first the primary focus of the study, with comparison to medical and others surgery management, including comparison with human medicine. Then discuss the other part. (move the part included between the lines 376-428 after the lines 317)
We strongly agree. In the discussion section the primary focus of the study was edited changing the lines 376-426 (corresponding to the revised article lines 331-385) after the lines 317 according to the previous Word document.
We appreciate the time you spent on revision, and we hope that the corrections will meet with approval.
Thank you very much and kind regards,
Michal Mol on behalf of the authors
